# Suppression Characteristics and Mechanism of Molasses Solution on Coal Dust: A Low-Cost and Environment-Friendly Suppression Method in Coal Mines

**DOI:** 10.3390/ijerph192416472

**Published:** 2022-12-08

**Authors:** Jianguo Liu, Tianyang Wang, Longzhe Jin, Gang Li, Shu Wang, Yixuan Wei, Shengnan Ou, Yapeng Wang, Jingge Xu, Minglei Lin, Jiahui Wang, Xianfeng Liu

**Affiliations:** 1NHC Key Laboratory for Engineering Control of Dust Hazard, University of Science and Technology Beijing, Beijing 100083, China; 2State Key Laboratory of Safety and Health for Metal Mine, Maanshan 243000, China; 3School of Civil and Resource Engineering, University of Science and Technology Beijing, Beijing 100083, China; 4College of Environment and Safety Engineering, Qingdao University of Science and Technology, Qingdao 266042, China; 5Institute of Occupational Health, China Academy of Safety Science and Technology, Beijing 100012, China; 6State Key Laboratory of Coal Mine Disaster Dynamics and Control, School of Resources and Safety Engineering, Chongqing University, Chongqing 400044, China

**Keywords:** coal dust, dust suppressant, molasses, wind erosion, moisturizing, agglutination, anti-evaporation

## Abstract

Coal dust pollution poses a serious public health threat. This study aimed to investigate the feasibility of creating a coal dust suppressant using molasses, a byproduct of the sugar industry. We studied the effects of a molasses solution of varying concentrations (i.e., ranging from 0% (pure water) to 40%) on the moisture, bonding, and wind erosion properties of coal dust. Overall, the effectiveness of the molasses increased with their concentration, and it manifested itself in the following way: (1) the molasses improved the anti-evaporation ability of wet coal dust. For example, the evaporation mass of the coal dust wetted using a molasses solution decreased by 82.8%; (2) molasses effectively agglutinated coal dust; (3) molasses can effectively decrease the surface tension and increase the viscosity of the wetting solution. The surface tension of the molasses solution reached 41.37 mN/m and the viscosity increased to 6.79 mPa·s; (4) molasses can significantly suppress the wind erosion of deposited coal dust, with its wind erosion mass decreasing 99.1%; finally, (5) the effectiveness of molasses at suppressing coal dust was discussed at a molecular level. This study highlights the feasibility of a low-cost and environment-friendly dust suppressant in coal mines.

## 1. Introduction

The control of coal dust is of great significance in ensuring the occupational health and safety of coal workers. The rapid development of science and technology has become synonymous with improved living standards for many parts of the world, and these improved living standards have been accompanied by an increase in energy demand [1,2,3]. With regard to energy, coal is one of the most widely used energy sources, and this is especially true in developing countries, where coal is the primary source of energy [4,5]. To meet the ever-increasing demands, the mining intensity of coal is increasing yearly, resulting in severe coal dust pollution in underground coal mines [6,7]. In recent years, more than 10,000 cases of pneumoconiosis have been recorded in China each year [8], and coal workers’ pneumoconiosis (CWP) has also been detected in developed countries such as Australia and the United States [9]. Therefore, research into cheaper means to suppress coal dust in underground mines is of great significance to ensure the health of coal workers across the globe.

Dust control technologies in underground coal mines can be divided into physical and chemical dust suppression [8]. Chemical dust suppression involves the addition of chemical materials into water, to alter the physicochemical properties of water. Such alterations improve the capture efficiency of water droplets on fugitive dust, resulting in the suppression of the further spreading of flying coal dust. Because of the hydrophobicity of most coal dust, the suppression efficiency of pure water solutions on coal dust is very low [8,10]. Therefore, chemical dust suppression has the potential to efficiently suppress dust in underground coal mines. However, owing to the current issues plaguing chemical dust suppression, this method has not been widely used in underground coal mines. The problems are mainly twofold: (1) high application costs; and (2) environmental pollution after the application. Therefore, it is worthwhile to develop chemical dust suppressants that are low-cost and environment-friendly [11,12].

In recent years, numerous studies have been conducted to develop low-cost and environment-friendly chemical dust suppressants. Using plant extraction technology, Zhang et al. [10] developed a highly permeable moistening additive for coal seam water injection, and a high dust reduction efficiency was achieved using the additive. Wang et al. [11] explored the wetting and moisturizing performances of glycerol microemulsions on coal seams. Liu et al. [13] prepared a modified dual-network dust suppression gel and efficiently suppressed flying coal dust. Wang et al. [14] compounded a green biodegradable dust suppressant using rhamnolipid, lactone sophorolipid, and surfactin and found that the compound showed significant wettability on coal dust. In addition, sodium polyacrylate, starch copolymer, and polymer guar gum are preferred materials for preparing environment-friendly coal dust suppressants [15,16,17]. However, from the abovementioned studies, we found the developed coal dust suppressants were primarily improved along with environmental friendliness. However, there were no improvements in terms of cost reduction, meaning that the developed suppressants cannot be widely applied in the mining industry. 

The use of industrial byproducts to prepare dust suppressants is an effective method for reducing the cost associated with developing chemical suppressants. Molasses is a by-product of sugar production and is produced worldwide [18,19]. Many studies have shown that molasses can effectively bond to particulate matter [19,20,21]. For example, Benk and Coban [22] showed that molasses can substitute cement as a binder material to produce lightweight and heat-insulating bricks. Zhong et al. [23] and Manyuchi et al. [24] found that the strength of coal briquettes was significantly improved when molasses was used as a binder. A similar result was reported by Kotta et al. [21], who found that the addition of molasses could significantly increase the strength of iron ore pellets. A study conducted by Huang et al. [19] showed that molasses could be utilized as a retarder for calcium sulfoaluminate cement-based mortars. Furthermore, focusing on the prominent binder properties of molasses, a few researchers have explored the possibility of using molasses as a rock dust suppressant. Gotosa et al. [25] studied the prevention characteristics of molasses on grave road dust and found that the molasses solutions decreased dust deposition by 77–83% compared to pure water. Parsakhoo et al. [12] found that molasses solutions have a significant bonding effect on particles with a size of less than 10 μm in forest roads. The study conducted by Omane et al. [26] showed that molasses can significantly decrease fugitive rock dust emissions on mine haul roads across various temperatures. In summary, the above-mentioned studies indicated that molasses can significantly suppress fugitive rock dust emissions because of its affinity to particulate matter. However, to the best of our knowledge, the ability of molasses to suppress coal dust (especially hydrophobic coal dust) has not been studied. Thus, it is important to determine if molasses is also an effective coal dust suppressant, as this would facilitate the development of low-cost and environment-friendly coal dust suppressants. 

The objective of this study was to explore the suppression characteristics of molasses solutions on hydrophobic coal dust (HCD). This objective was achieved in three ways: (1) the moisturizing and agglutination performance of molasses solution on the HCD was determined; (2) the properties of the molasses solution were tested with various concentrations; and (3) the dust suppression efficiency of molasses on the HCD was measured using wind erosion experiment. Finally, the suppression mechanism of molasses solution on coal dust was discussed, and methods were proposed to improve the preparation of molasses-based coal dust suppressants. This study is significant for the development of low-cost and environment-friendly dust suppressants in coal mines.

## 2. Material and Methods

### 2.1. Material

(1)Molasses

The sugarcane molasses used in this study were purchased from Jinqianwan Molasses Co., Ltd. (located in Liuzhou City, Guangxi Province, China). The composition and content of sugarcane molasses are listed in Table 1 [18,19]. 

(2)Coal dust samples

Following GB/T 482-2008, the coal explored in this study was sampled from an underground mining site located in Datong City, Shanxi Province, China. The proximate contents of the coal samples were measured according to GB/T 212-2008. The maceral composition and vitrinite reflectance (R_o_) were tested following ISO 7404-1:2016 and ISO 7404-1:2016, respectively. The results are presented in Table 2. 

### 2.2. Experimental Setup

#### 2.2.1. Overview

As shown in Figure 1, this study includes three main approaches. First, the moisturizing and agglutination performances of molasses solutions in various concentrations on coal dust were explored. Secondly, four physical parameters of the molasses solutions were tested at different concentrations. Third, the dust suppression efficiency of the molasses solution on coal dust was measured. Combining insights from the first and second research approaches, the suppression mechanism of molasses solution on coal dust was analyzed, and the improved methods for preparing molasses-based coal dust suppressants were proposed. 

#### 2.2.2. Moisturizing and Agglutination Performance Tests

##### Moisturizing Performance Test

Moisturization is an important property of dust suppressants, and it largely affects dust suppression performance [7,11]. To explore the effect of molasses on the moisture characteristics of coal dust, the evaporation parameters of mixed wet coal dust (MWCD) and moisturizing parameters of mixed dry coal dust (MDCD) were measured. The specific tests are described below.

(1)Evaporation parameters of the MWCD

First, the coal blocks sampled from the underground mining site were broken and milled using a ball mill machine. Coal particles with a size of less than 75 μm (200 mesh) were then screened. The coal dust was then mixed with the molasses solutions (with mass concentrations of 0%, 5%, 10%, 15%, 20%, 30%, and 40%) at a 2:1 mass ratio. The two elements (i.e., coal dust and molasses solution) were stirred slowly to ensure full mixing. Once this was complete, the MWCD sample was obtained. The evaporation parameters of the MWCD were measured as follows: *M* g of MWCD was placed into a glass dish with an inner diameter of 8.0 cm (the mass of the dish is *X* g); next, the dish was exposed to a room temperature condition (Beijing area, autumn, the temperature is 25 ± 1℃, relative humidity is 40–45%); the mass of the dish was then tested per hour and recorded as *M_n_* (*n* presents time, h), and the moisture evaporation mass could then be calculated following Equation (1). Subsequently, taking the time as the abscissa and the evaporation mass as the ordinate for drawing, the data were linearly fitted using the least square method, and the slope of the fitted linear is the evaporation rate (g/h).
(1)$$m_{n}=M-\left(M_{n}-X\right)$$
where *m_n_* is the evaporation mass of the MWCD at time *n* (g); *M* is the mass of the MWCD before evaporation (g); *M_n_* is the total mass of the MWCD and dish (g), and *X* is the mass of the dish (g). 

(2)Moisturizing rate test for MDCD.

The MWCD was placed into a plastic dish with an inner diameter of 8 cm. The dish was then placed into an oven at a temperature of 100 ℃. The mass of the dish was measured per hour, and the mixed dry coal dust (MDCD) was obtained when the mass no longer changed. The plastic dish was then removed and the MDCD was placed into an oven, in which the temperature and relative humidity could be adjusted. The moisturizing parameters of the MDCD were measured at a temperature of 25 ℃ and various relative humidities (70%, 80%, and 90%). The mass of the dish was measured every 30 min for 4 h; after that, the maximum moisturizing mass (g) and moisturizing rate parameter were calculated according to the Weibull mode following Equations (2) and (3) [27]: (2)Mt=M0+(M∞−M0)×[1−e(−tβ)]
(3)Mt=M∞×[1−e(−tβ)]
where *M_t_* is the mass of MDCD at time *t* (g); *M*_0_ is the mass of MDCD before moisturizing (g); *M_∞_* is the maximum moisturizing mass of MDCD (g/g); and *β* is the evaporation rate parameter of MDCD. 

##### Agglutination Performance Test

In this study, to evaluate the agglutination performance of molasses solutions on coal dust, the maximum compressive strength of an MDCD block released from a plastic dish was tested, as shown in Figure 2. We believe that the greater the maximum compressive strength, the stronger the bonding effect of the solution on coal dust. As depicted in Figure 2c, a pressure meter vertically compresses the MDCD block at its center point. After the destruction of the MDCD block, as shown in Figure 2d, the maximum value recorded in the pressure meter is the maximum compressive strength of the MDCD block. The pressure surface of the pressure meter was a cylinder with a diameter of 2 mm. Using this method, the agglutination performance of the molasses on coal dust was evaluated across various concentrations (0%, 5%, 10%, 15%, 20%, 30%, and 40%). 

#### 2.2.3. Physical Properties Tests of Molasses Solution

(1)Surface tension tests

Surface tension has significant effects on both the evaporation characteristics and the wettability of the liquid [28]. The pendant drop method [29] was used in this study to measure the surface tension of molasses solutions with various concentrations (i.e., 0%, 5%, 10%, 15%, 20%, 30%, and 40%) in a Theta Lite TL 101 (Finland) apparatus [7]. 

(2)Contact angle tests

Wettability is an important parameter for evaluating the performance of coal dust suppressants because most coal dust is hydrophobic [8,30,31], and the contact angle is commonly used to evaluate the wettability of a solution on coal dust [6,30]. Therefore, in the present study, the contact angles between molasses solutions with concentrations of 0%, 5%, 10%, 15%, 20%, 30%, and 40% and coal dust were measured using the Theta Lite TL 101 (Finland). This test has been described in detail in our previous study [32]. 

(3)Viscosity tests

Viscosity is a key parameter that affects the agglutination and fluidity properties of a solution [33,34], thus significantly influencing the performance of dust suppressants. An RV-SSR viscosimeter (China) was used to measure the viscosity of molasses solutions of varying concentrations (i.e., 0%, 5%, 10%, 15%, 20%, 30%, and 40%).

(4)Evaporation rate tests

The evaporation rate of the solution determines whether the solution can sustainably moisturize a substance [35,36]. Therefore, the evaporation rates of molasses solutions with concentrations of 0%, 5%, 10%, 15%, 20%, 30%, and 40% were tested at room temperature. The method used was the same as that used in the measurement of the evaporation rate of MWCD (i.e., described in “Moisturizing Performance Test” section). 

#### 2.2.4. Dust Suppression Efficiency Measurement

According to [26], the suppression efficiency of molasses solutions on coal dust was measured using the method illustrated in Figure 3. First, to simulate the surface materials of the underground coal roadway, fine coal dust with a particle size of less than 75 μm was fully mixed with large coal particles with a particle size larger than 2.0 mm at a mass ratio of 1:9, and 200 ± 1 g of the mixed coal particles was evenly placed on a square acrylic board with a side length of 20 cm, after which 10 mL of molasses solution was symmetrically sprayed on the surface of the mixed coal particles; after that, the sprayed samples were placed at a room temperature for 7 days (Figure 3a). Finally, the wind erosion test of the prepared samples was conducted in a closed chamber (Figure 3b). 

For the wind erosion test, a fan was placed horizontally to the test sample, and the wind speed at the surface of the test sample was 5 m/s (the maximum wind speed in underground return roadway). This speed was controlled by adjusting the distance between the fan and the test sample. The test time was 5 min, and the mass of the test sample was measured before and after the wind erosion test, and the amount of airborne particulate matter (PM) with a particle size less than 10 μm (PM10) was counted during the test at 1 m behind the test sample using a particulate matter quantity tester (FULKE 985). From this test, the wind erosion mass (Δm) and the cumulative number of PM10 particles were obtained to characterize the suppression efficiency of the molasses solutions on coal dust. 

(1)Wind erosion mass

The wind erosion mass (Δm) is a quantity that represents the mass of the test sample before and after the wind erosion test. The quantity can be calculated as follows: (4)$$\Delta m=m_{0}-m_{1}$$
where Δm is the wind erosion mass (g); and *m*_0_ and *m*_1_ are the masses of the test sample before and after the wind erosion test (g), respectively.

To analyze the suppression efficiency of molasses solution on the test samples, the Δm reduction percentage (P) of the molasses solution over that of pure water (0% molasses) was computed using Equation (5).
(5)$$P_{n}=\frac{\Delta m_{n}-\Delta m_{w}}{\Delta m_{w}}\times 100\%$$
where *P_n_* is the Δm reduction percentage of the molasses solution with a concentration of *n* (%) over that of pure water without molasses (%); and Δm*_n_* and Δm_w_ are the wind erosion mass (g) of the molasses solution with a concentration of *n* and pure water, respectively. 

(2)Cumulative number of PM10 particles.

The FULKE 985 apparatus used in the wind erosion test can measure the number of particles in various sizes (i.e., 0.3, 0.5, 1.0, 2.0, 5.0, and 10.0 μm). To characterize the suppression efficiency of molasses solutions on coal dust, the relative percentage of the cumulative number of PM10 particles between the molasses solution and pure water was calculated using Equation (6).
(6)$$C_{n}=\frac{c_{w}-c_{n}}{c_{w}}\times 100\%$$
where *C_n_* is the relative percentage of the cumulative number of PM10 in the test sample sprayed with the molasses solution with a concentration of *n* compared with that of the test sample sprayed with pure water (%); and c_w_ and c*_n_* are the cumulative number of PM10 in the test sample sprayed with pure water and the molasses solution with the concentration of *n*, respectively. 

Note that the cumulative number of PM10 particles is the total number of particles with sizes of 0.3, 0.5, 1.0, 2.0, 5.0, and 10.0 μm. It can be computed using Equation (7).
(7)$$c=\sum \mathrm{pm}{=\mathrm{pm}}_{0.3}+{\mathrm{pm}}_{0.5}+{\mathrm{pm}}_{1.0}+{\mathrm{pm}}_{2.0}+{\mathrm{pm}}_{5.0}+{\mathrm{pm}}_{10.0}$$
where c is the cumulative number of PM10 particles; and pm*_i_* is the number of particles with a particle size of *i* (*i* = 0.3, 0.5, 1.0, 2.0, 5.0, 10.0). 

## 3. Results

### 3.1. Moisturizing Performance of Molasses on the Coal Dust

(1)Evaporation parameters of the MWCD

The evaporation parameters of the MWCD, including the evaporation mass and evaporation rate, were measured, and the results are shown in Figure 4. It can be observed from Figure 4a, at the same evaporation time, the evaporation mass of the MWCD sample gradually decreases with the increase in molasses concentration; in 3.5 h, the evaporation mass of the MWCD wetted by 40% molasses solution is only 27.2% of that of the MWCD wetted by pure water. This result confirms that molasses has a significant effect on the anti-evaporation ability of coal dust. 

Figure 4b depicts the variation in the evaporation rate of the MWCD with molasses concentration. Specifically, as molasses concentration increases, the evaporation rate of the MWCD can be divided into three stages. In stage 1, there was no significant change in the evaporation rate of the MWCD, where the molasses concentration is less than 10%; this indicates that the molasses did not affect the evaporation of coal dust when their concentration is less than 10%. In stage 2, the evaporation rate of the MWCD rapidly decreases with the increase in the molasses concentration from 10% to 30%; the evaporation rate of the MWCD at the 30% molasses concentration is 30.5% of that at 10% molasses concentration. In stage 3, with the further increase in molasses concentration, the evaporation rate of the MWCD at 30% molasses concentration slightly reduced from 0.064 to 0.056 g/h at 40% molasses concentration. The result shows that the molasses could significantly improve the anti-evaporation ability of coal dust when its concentration is greater than 10%. Moreover, the concentration-dependent effectiveness increases until the concentration of molasses is greater than 30%. 

(2)Moisturizing characteristics of the MDCD

The moisture characteristics of the MDCD were measured at various relative humidity values, and the results of the moisturizing mass variation with time are shown in Figure 5. According to the moisturizing mass and the Weibull model [27], the maximum moisturizing mass and moisturizing rate of the MDCD sample were obtained, as shown in Figure 6a and Figure 6b, respectively. 

It can be observed from Figure 6a that the molasses concentration has a significant effect on the maximum moisturizing mass of the coal dust, and the effect largely depends on the relative humidity. Specifically, first, the maximum moisturizing mass of the MDCD sample exponentially increases with the rise in molasses concentration with a goodness-of-fit (R^2^) large than 0.90 in three relative humidities; second, in higher relative humidity, the molasses make a more apparent improvement in terms of their maximum moisturizing mass of coal dust. At 70% relative humidity, molasses had a limited effect on the maximum moisturizing mass of coal dust at concentrations less than 30%, whereas, at 90% relative humidity, molasses showed an apparent improvement in the maximum moisturizing mass of coal dust at low concentrations. Therefore, it can be concluded that molasses has a more significant improvement in the moisturizing ability of coal dust in a space with higher relative humidity. 

Within the context of suppressants, the moisturizing rate is a key parameter for assessing the performance of a material. As shown in Figure 6b, the concentration of the molasses solution has a significant positive effect on the moisturizing rate of the MDCD; that is, with the increase in molasses concentration, the moisturizing rate of the MDCD rapidly increases first and then slowly increases. In addition, the variation in the moisturizing rate was consistent under different relative humidity values, indicating that the relative humidity had a limited influence on the moisturizing rate of the MDCD. 

In summary, the concentration of the molasses solution has a significant effect on both the maximum moisturizing mass and the moisturizing rate of the MDCD, and the relative humidity of the surrounding environment can improve the maximum moisturizing mass of the MDCD and had a limited influence on the moisturizing rate of the MDCD. 

### 3.2. Agglutination Performance of Molasses on the Coal Dust

The agglutination performance of a solution is highly related to the efficiency of the secondary flying of coal dust [16]. Therefore, in this section, the agglutination performance of the molasses solution on the coal dust with various concentrations is tested, and the results are depicted in Figure 7. 

The maximum pressure of the coal dust block (Figure 7b) was measured and the results are shown in Figure 8. It was observed that molasses had a significant positive effect on the maximum pressure of the coal dust block. Specifically, pure water (0% molasses concentration) has almost no bonding effect on coal dust, and the maximum pressure is only 1.15 N. However, the maximum pressure exponentially rises with the increase in the molasses concentration with an R^2^ = 0.99, and at 40% molasses concentration, the maximum pressure reached 171.21 N, which is 148.9 times that of pure water. These results indicate that the molasses solution has a significant agglutination effect on the coal dust. Therefore, molasses could be applicable in the field in terms of suppressing secondary-flying dust in coal mine roadways.

### 3.3. Physical Properties of Molasses Solution

To fully understand the moisturizing and agglutination functions of molasses solutions on coal dust, the physical properties of molasses solutions with varying concentrations were measured, including surface tension, wettability, viscosity, and evaporation parameters. These results are detailly described in the following subsection. 

(1)Surface tension and wettability of molasses solutions

Figure 9a depicts the variation in the surface tension of the molasses solution with the concentration. It can be observed that molasses can significantly decrease the surface tension of the solution. Specifically, the surface tension exponentially decreased following the rise of molasses concentration with an R^2^ = 0.92; the surface tension decreased to 41.37 mN/m at 40% molasses concentration, which reduced by 43.07% lower than that of pure water (0% molasses concentration). Owing to the hydrophobicity of most coal dust [6], reducing the surface tension can improve the wettability of the solution to coal dust. In this regard, molasses points to being an effective coal dust suppressant. 

Figure 9b depicts the contact angle variation between the molasses solution and the coal dust. It was found that with the increase in molasses concentration, the contact angle underwent two variation stages. In stage 1, the contact angle decreased with an increase in the molasses concentration when the molasses concentration was less than 20%; subsequently, with a further increase in the concentration, the contact angle began to increase slightly. The minimum value of the contact angle was 112.38° achieved at 20% molasses concentration. According to [37], the solution cannot wet the coal dust when the contact angle is larger than 90°. Based on these results, it was determined that although molasses can decrease the solution surface tension, it cannot wet hydrophobic coal dust. Therefore, enhancing the wettability of molasses solutions will be a key step in the preparation of molasses-based dust suppressants. 

(2)Viscosity of molasses solutions

Viscosity is an important parameter that affects the fluidity and spray properties of liquid [38]. The concentration-dependent variation in the viscosity of molasses solutions is shown in Figure 10. The viscosity of molasses solution exponentially increases with the increase in the molasses concentration with an R^2^ = 0.99; at 40% concentration, the viscosity of molasses solution reached 6.79 mPa·s, which is 5.43 times that of pure water. That is, adding molasses to water can increase the viscosity of the solution, which reduces its fluidity and spray properties. Therefore, the concentration should be controlled when preparing molasses-based coal dust suppressants to reduce their influence on solution viscosity. 

(3)Evaporation parameters of molasses solutions

The anti-evaporation characteristics of the dust suppressant solution are the key reasons for the enhanced water retention of dust. The evaporation parameters of the molasses solutions at room temperature are shown in Figure 11. As depicted in Figure 11a, the evaporation mass of molasses solution is tightly related to its concentration; at the evaporation time of 3 h, the maximum evaporation mass of molasses solution was 13.76 g, which was reached at 10% concentration. The minimum evaporation mass was 7.83 g, reached at 40% concentration, and the evaporation mass of the pure water was 11.61 g. That is, molasses can both increase and decrease the evaporation mass of the solution, which depends closely on its concentration.

To further analyze the effect of molasses concentration on the evaporation properties of the solution, as shown in Figure 11b, the evaporation rates of the molasses solutions with different concentrations over 3 h were computed. The results show that with the increase in molasses concentration, the evaporation rate of the solution increases first and decreases then, and a turning point occurs at 10% concentration. The evaporation rate at 40% concentration reached a minimum value of 2.73 g/h, which was 29.82% lower than that of pure water. The results show that the effect of molasses on the solution evaporation rate is heavily concentration-dependent; that is, it can improve the evaporation rate at low concentrations and decrease the evaporation rate at high concentrations. 

### 3.4. Suppression Efficiency of Molasses on the Coal Dust

To investigate the suppression performance of molasses solution on the secondary flying of coal dust, in this section, a wind erosion test is conducted for coal dust sprayed with molasses solutions. Figure 12 shows the variation in the wind erosion mass (Δm) and its reduction percentage (P) compared with pure water. We saw that Δm exponentially decreases following the increase in molasses concentration with an R^2^ = 0.95, and the Δm values of the coal dust samples sprayed with 20% and 40% concentration molasses are 0.22 g and 0.02 g, which reduced 90.0% and 99.1% compared with that of pure water, respectively. This result indicates that molasses has a significant suppression effect on the wind erosion of coal dust, and its suppression efficiency largely depends on the molasses concentration. 

Figure 13a depicts the number of flying particles with different particle sizes in the wind erosion test. It can be seen that, first, compared with the coal dust sample sprayed with pure water, the number of flying particles in the coal dust samples sprayed with molasses solution largely decreased; secondly, the particle sizes of the flying particles from the coal dust sample sprayed with pure water were mostly less than 2.0 μm; however, the particle sizes of the flying particles from the coal dust samples sprayed with molasses solution were mostly approximately 0.3 μm, and the number of particles with larger particle size was also significantly less than that from the pure water-sprayed coal dust sample. This result also points to an apparent suppression effect of molasses solution on the wind erosion of coal dust compared with pure water. 

Figure 13b presents the variation in the relative percentage of PM10. It can be seen that, similar to the variation in the relative percentage of molasses solution shown in Figure 12, the relative percentage of the PM10 number exponentially increases with the increase in the molasses concentration with an R^2^ = 0.99. The PM10 number of the coal dust sample sprayed with 5% and 40% molasses solutions was reduced by 76.9% and 91.2%, respectively, compared with that of the coal dust sprayed with pure water. This result well verified the data shown in Figure 12. In summary, the molasses solution exhibited good dust suppression performance in the wind erosion test, indicating its feasibility in acting as a coal dust suppressant is high.

### 3.5. Relationships of the Moisturizing and Agglutination Parameters with the Dust Suppression Efficiency

To reveal the effect of the moisturizing and agglutination parameters on the dust suppression efficiency, the wind erosion mass was used as the dust suppression parameter, and the relationships between the evaporation rate of the MWCD, the maximum moisturizing mass of the MDCD, and a maximum pressure of the agglutination coal block with the wind erosion mass were established and analyzed in this section. 

#### 3.5.1. Relationships between Moisturizing Parameters and the Wind Erosion Mass

Figure 14 presents the relationships between the evaporation rate of the MWCD, the maximum moisturizing mass of the MDCD, and wind erosion mass. It can be seen that with an increase in the evaporation rate and maximum moisturizing mass, the wind erosion mass exponentially increases with R^2^ = 0.77 and exponentially decreases with R^2^ = 0.99, respectively. That is, improving the moisturizing ability of coal dust has a positive effect on preventing wind erosion and the secondary flying of coal dust, but the effect is limited. Specifically, after the moisturizing performance of coal dust reaches a certain point, the suppression effect will not be improved by further enhancing the moisture ability. 

#### 3.5.2. Relationship between Agglutination Parameter and the Wind Erosion Mass

Figure 15 shows the relationship between the maximum pressure of the agglutination coal dust block and the wind erosion mass. It was found that the wind erosion mass linearly decreased with an increase in the maximum pressure with R^2^ = 0.96. This result indicates that the agglutination effect could directly affect the wind erosion resistance of coal dust, and the higher the maximum pressure, the stronger the bonding effect and the more significant the dust suppression efficiency. 

Based on the above results, we can conclude that the suppression effect of molasses on coal dust can be primarily attributed to enhancing the moisturizing and bonding properties of coal dust, and the bonding effect is the dominant factor. 

## 4. Discussion

In this study, the feasibility of using molasses as a coal dust suppressant was evaluated through the lens of experiments. The results showed that molasses can significantly improve the moisturizing ability of coal dust and strongly bond coal particles. Through these two effects, the molasses solution has a remarkable suppression effect on the secondary flying of coal dust caused by wind erosion. In this section, we place our results within the context of published literature and discuss their implications.

### 4.1. Wetting and Moisturizing Functions of Molasses on Hydrophobic Coal Dust

Molasses contain a lot of carbohydrates, such as sucrose, glucose, and fructose, among which the sucrose content can reach approximately 48.8% [18]. Sucrose molecules contain large amounts of functional groups containing oxygen elements such as hydroxyl and ether, as shown in Figure 16a, these functional groups are hydrophilic. Additionally, the sucrose molecules also contain two hydrophobic hexane chains. In other words, sucrose molecules contain both hydrophilic and hydrophobic groups, which could function as surfactants. Therefore, molasses could effectively reduce the surface tension of water (Figure 9a). However, the molecular weight of the hydrophilic group in the sucrose molecule is comparable to that of the hydrophobic group, which limits the ability of molasses to reduce the surface tension of water. Figure 9a validates this analysis, and the surface tension of the molasses solution with 40% concentration was still larger than 40 m·N/m. According to a study by Xu et al. [39], when the solution surface tension is greater than 45 m N/m, it cannot wet the hydrophobic coal dust. Our findings are congruent with this conclusion (Figure 9b); the minimum contact angle between the molasses solution and the hydrophobic coal dust was greater than 90°, indicating that the molasses solution cannot effectively wet the hydrophobic coal dust. Therefore, a small amount of surfactant would need to be added to improve the wettability of the molasses solution during the development of a molasses-based coal dust suppressant.

In addition, molasses also contains plenty of ash (approximately 13%) [18], which can increase the solid content and viscosity of the solution. Furthermore, as shown in Figure 16b, when molasses is dissolved in water, the hydrogen in the hydroxyl group of the sucrose molecule can quickly form a hydrogen bond with the oxygen element in the water molecule, and the oxygen in the ether group of the sucrose molecule can also form hydrogen bonds with the hydrogen in the water molecule. The viscosity of the molasses solution increases under the action of hydrogen bonding. Therefore, as presented in Figure 10, the viscosity of the molasses solution exponentially increases with an increase in molasses concentration. With an increase in the concentration of molasses, the solution surface tension did not change (Figure 9a), but its viscosity increased rapidly (Figure 10), which may be the reason why the contact angle increased again when the molasses concentration was greater than 20% (Figure 9b).

In addition, molasses contains a few metal ions, such as Ca, Mg, Na, and K, and trace chlorides [18], and these elements easily form compounds such as CaCl_2_ and MgCl_2_ after water evaporation. CaCl_2_ and MgCl_2_ can form CaCl_2_·*x*H_2_O and MgCl_2_·*x*H_2_O, respectively, by absorbing moisture in the air [7,40]. Therefore, mixed wetting coal dust (MWCD) and mixed dry coal dust (MDCD) exhibit significant anti-evaporation characteristics (Figure 4) and moisturizing properties (Figure 6), respectively. 

### 4.2. Agglutination Molasses on Hydrophobic Coal Dust

Molasses solutions exhibit significant agglutination properties for coal dust. As shown in Figure 17, after spraying the molasses solution on the coal dust surface, a bonding layer (Figure 17d) can be formed on the surface of the deposited coal dust after drying, which is the main reason why the molasses solution has a significant wind erosion suppression effect on the deposited coal dust (Figure 15).

As shown in Figure 16c, when the molasses solution was mixed with the coal dust, the sucrose molecules in the solution contacted the coal dust surface. It is well known that coal contains large amounts of organic polymers mainly composed of carbon, hydrogen, and oxygen. Thus, on the surface of coal dust, there is a large number of alkyl, alkenyl, and other groups [41,42,43], and the hydrogen and oxygen in these groups can rapidly form hydrogen bonds with the oxygen and hydrogen in sucrose and water. Using hydrogen bonds, coal dust particles, sugar molecules, and water molecules could quickly form physical networks, resulting in the macroscopic phenomenon of the bonding of coal dust particles, as shown in Figure 17d. Therefore, an agglutination layer can be formed when spraying a molasses solution on the surface of deposited coal dust, such as in open coal piles and underground roadways. The agglutination layer can effectively prevent wind erosion of the deposited coal dust, producing a dust suppression effect. 

In summary, in this study, we found that molasses could effectively improve the moisturizing and agglutination properties of coal dust. Under the action of these two characteristics, the molasses solution has significant wind erosion resistance to the deposited coal dust. At the same time, molasses solutions have some disadvantages in preventing coal dust, such as poor wettability and high viscosity. Therefore, the wettability and viscosity should be fully improved when developing a coal dust suppressant using molasses. 

## 5. Conclusions

The objective of this study was to investigate the suppression performance and mechanism of molasses solutions on coal dust at various concentrations. To achieve this objective, the moisturizing and agglutination of molasses on coal dust was tested, the physical properties of molasses solutions with different concentrations were measured, and the dust suppression efficiency of molasses on the deposited coal dust was investigated. Based on the above experiments, the suppression mechanism of molasses on coal dust is discussed, and the advantages and disadvantages of molasses in facilitating the creation of coal dust suppressants are analyzed. The following conclusions can be drawn: (1)Molasses can improve the anti-evaporation ability of wet coal dust. The evaporation mass of the MWCD experienced three stages as the molasses concentration increased from 0% to 40%, and the evaporation mass decreased by 82.8% at 40% concentration than that at 0% concentration (pure water). (2)Molasses can enhance the moisturizing properties of coal dust. The moisturizing mass of the MDCD exponentially increased with an increase in molasses concentration, but the moisturizing rate was slightly influenced by the molasses concentration. (3)Molasses has a significant agglutination effect on coal dust. The bonding pressure of molasses solution on coal dust exponentially increases with the increasing of molasses concentration, and the maximum pressure of the coal dust block bonding by 40% molasses solution reached 171.21 N, which is 148.9 times that of the coal dust block bonding by pure water. (4)Molasses can effectively decrease the surface tension of a solution and increase its viscosity. At 40% concentration, the surface tension of the molasses solution reached 41.37 mN/m and the viscosity increased to 6.79 mPa·s. (5)Molasses can significantly suppress the wind erosion of deposited coal dust. The wind erosion mass of the deposited coal dust exponentially decreased with an increase in molasses concentration, and the wind erosion mass decreased by 99.1% at 40% concentration than that at 0% concentration (pure water).(6)The use of molasses to create a coal dust suppressant is highly feasible. However, the wettability of molasses solutions should be improved, and the viscosity of molasses solutions should be considered when developing molasses-based coal dust suppressants. 

## Figures and Tables

**Figure 1 ijerph-19-16472-f001:**
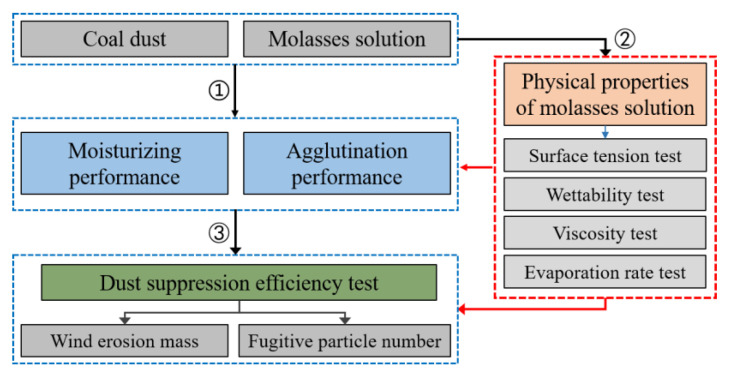
Research flow chart of the present study.

**Figure 2 ijerph-19-16472-f002:**
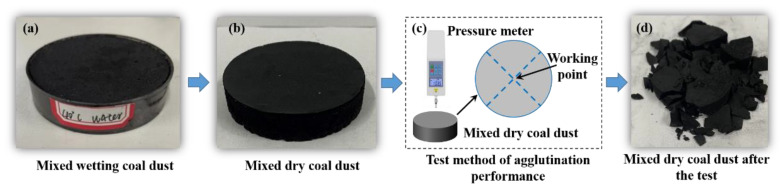
Testing process of the agglutination performance of molasses solutions on coal dust: (**a**) mixed wetting coal dust, (**b**) mixed dry coal dust, (**c**) test method of agglutination performance, and (**d**) mixed dry coal dust after the test.

**Figure 3 ijerph-19-16472-f003:**
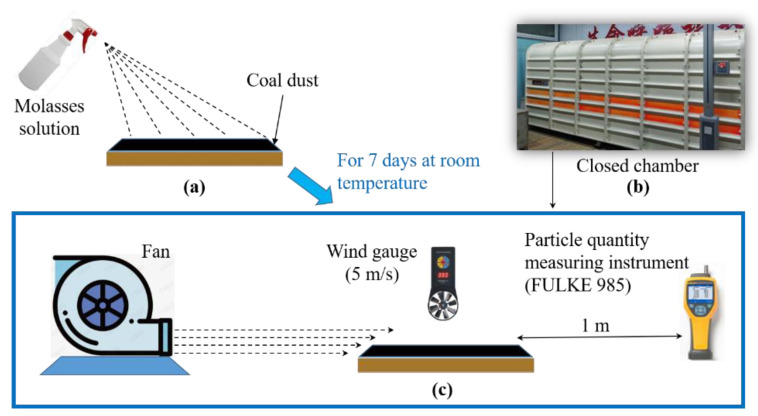
Method for measuring the suppression efficiency of molasses solution on coal dust: (**a**) spray process, (**b**) the closed chamber, and (**c**) wind erosion test.

**Figure 4 ijerph-19-16472-f004:**
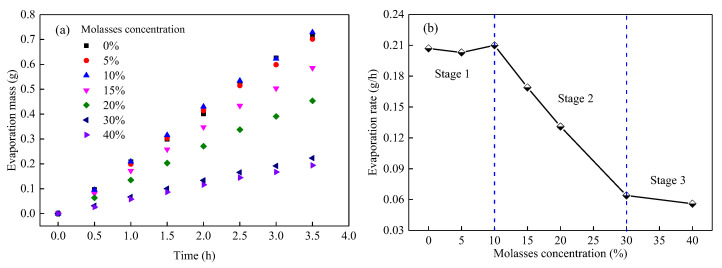
The evaporation mass (**a**) and evaporation rate (**b**) of the mixed wetting coal dust with various molasses concentrations.

**Figure 5 ijerph-19-16472-f005:**
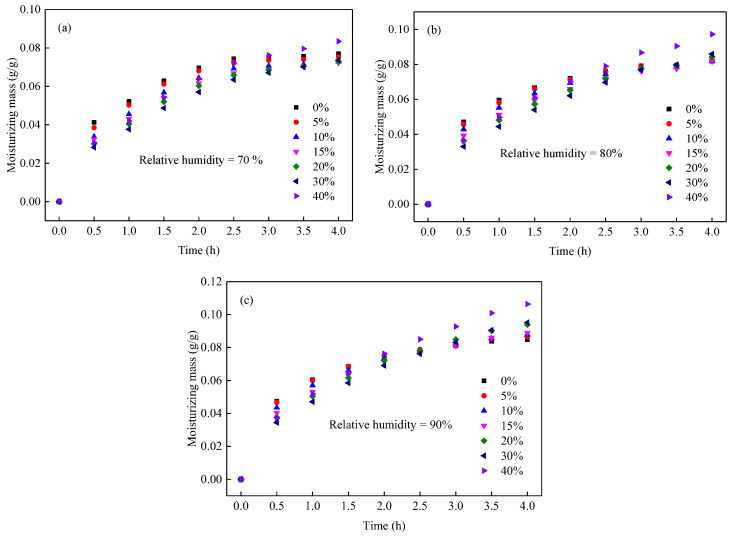
The moisturizing mass of mixed dry coal dust at the relative humidity of 70% (**a**), 80% (**b**), and 90% (**c**).

**Figure 6 ijerph-19-16472-f006:**
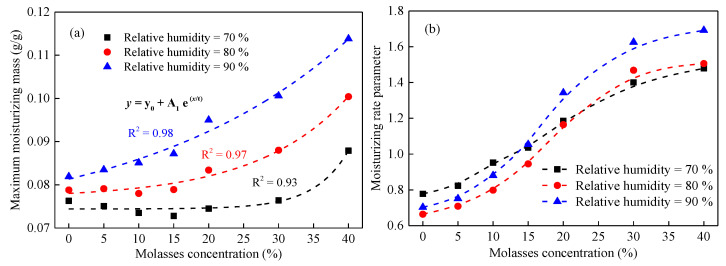
The variation in maximum moisturizing mass (**a**) and moisturizing rate parameter (**b**) of mixed dry coal dust with the concentration of molasses.

**Figure 7 ijerph-19-16472-f007:**
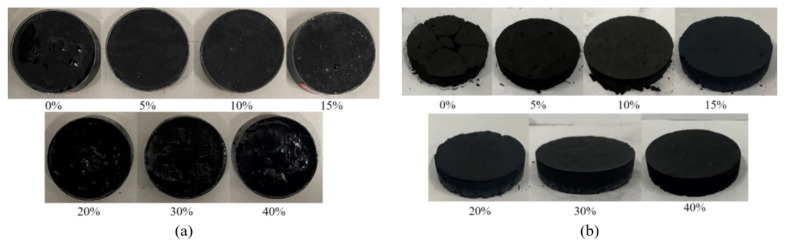
The mixed wetting coal dust (**a**) and the mixed dry coal dust blocks (**b**) with various molasses concentrations.

**Figure 8 ijerph-19-16472-f008:**
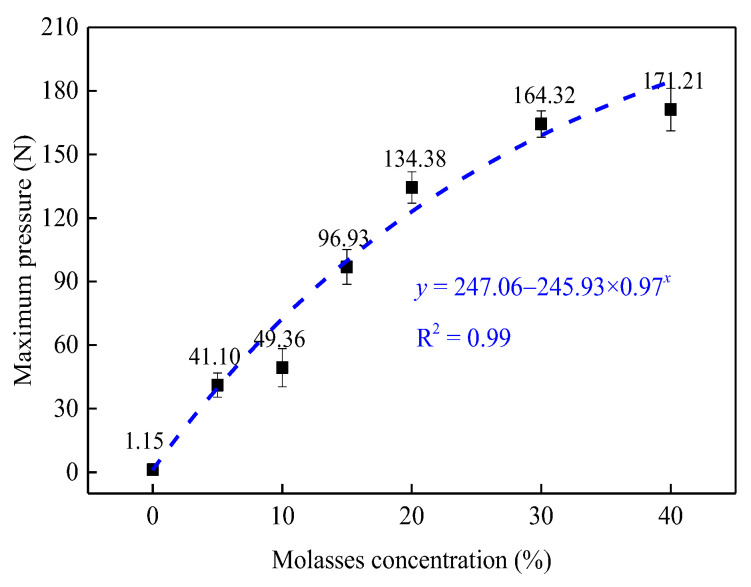
Maximum pressure variation of mixed dry coal dust blocks with molasses concentration.

**Figure 9 ijerph-19-16472-f009:**
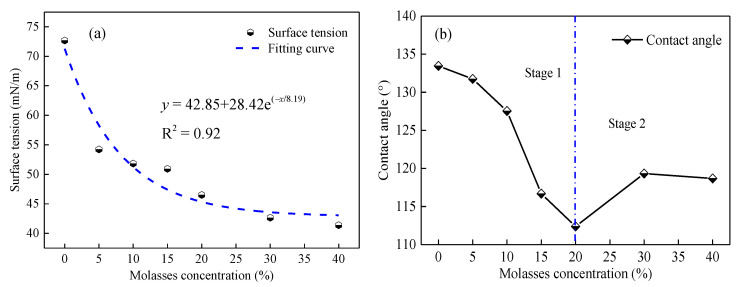
The surface tension of molasses solution (**a**) and the contact angle between coal dust sample and molasses solution (**b**) under various molasses concentrations.

**Figure 10 ijerph-19-16472-f010:**
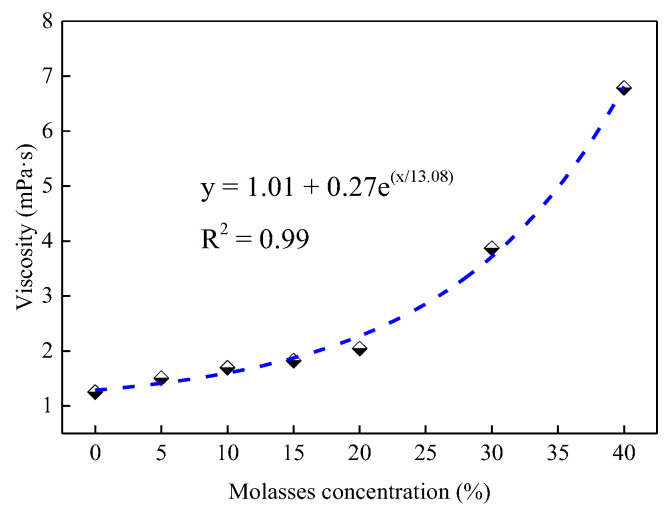
The viscosity of molasses solution with various concentrations.

**Figure 11 ijerph-19-16472-f011:**
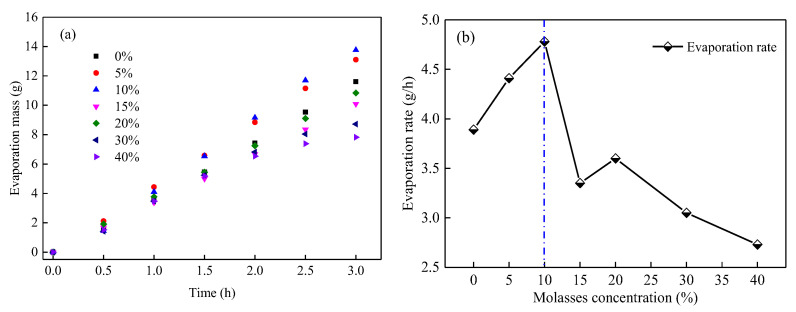
Evaporation mass (**a**) and evaporation rate (**b**) of molasses solution with various concentrations.

**Figure 12 ijerph-19-16472-f012:**
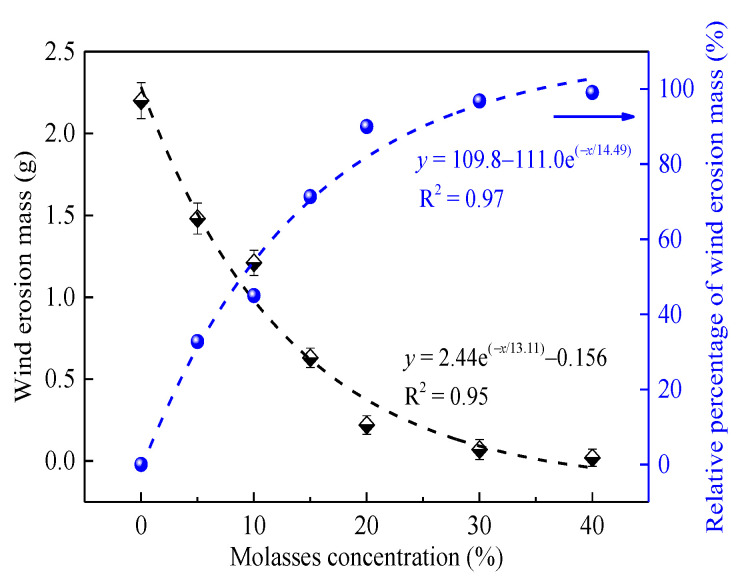
Variation in the wind erosion mass and its relative percentage of molasses solution with concentration.

**Figure 13 ijerph-19-16472-f013:**
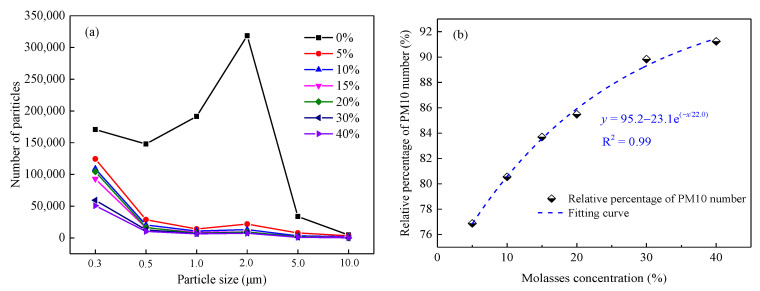
Variation in particle number (**a**) and relative percentage of PM10 number (**b**) of molasses solutions with concentration.

**Figure 14 ijerph-19-16472-f014:**
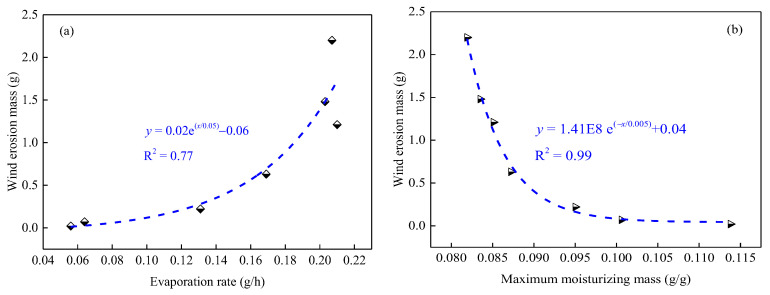
Relationships between the evaporation rate of the wetting mix coal dust and the wind erosion mass (**a**), and between the maximum moisturizing mass of the dry mixed coal dust and the wind erosion mass (**b**).

**Figure 15 ijerph-19-16472-f015:**
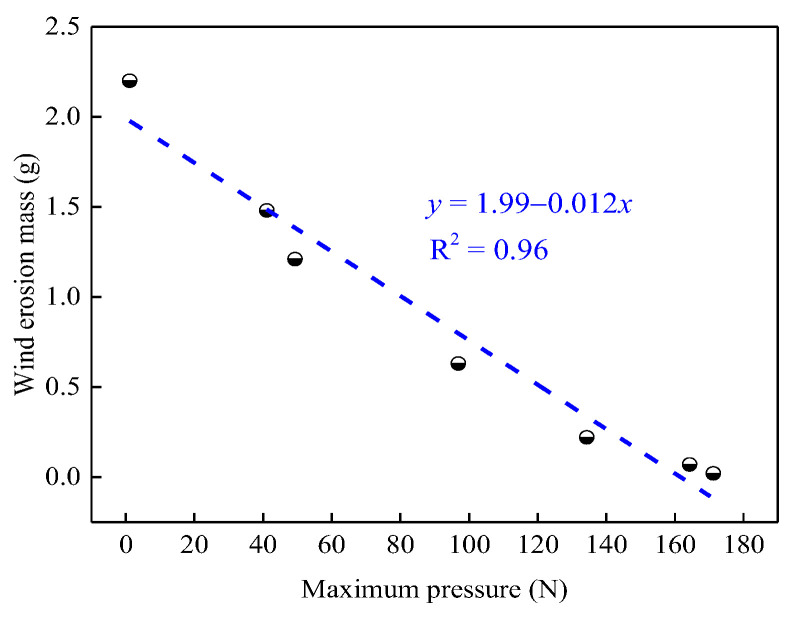
Relationship between the maximum pressure of the dry mixed coal dust block and the wind erosion mass.

**Figure 16 ijerph-19-16472-f016:**
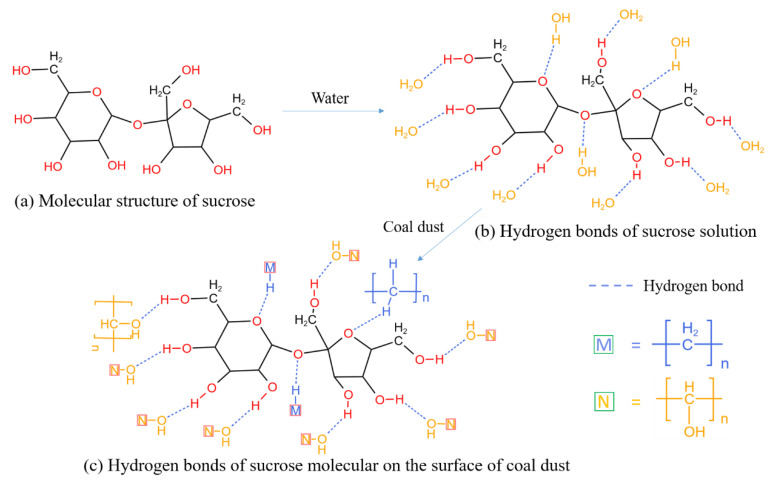
The molecular structure of sucrose (**a**), the hydrogen bonds of sucrose molecular in water (**b**), and the hydrogen bonds of sucrose molecular on the surface of coal dust (**c**).

**Figure 17 ijerph-19-16472-f017:**
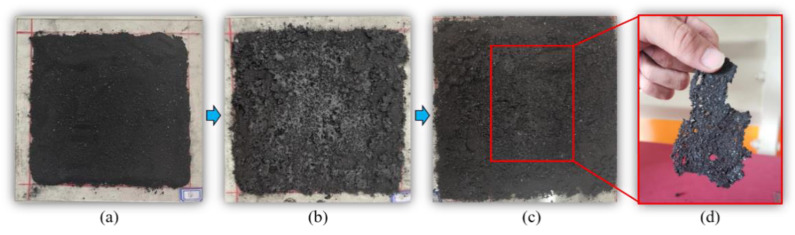
Agglutination of molasses solution on the hydrophobic coal dust: dry coal dust (**a**), wetting coal dust by molasses solution (**b**), dried coal dust mixed with molasses (**c**), and the agglutination of molasses on coal dust (**d**).

**Table 1 ijerph-19-16472-t001:** The compositions of the molasses [18,19].

Number	Main Constituents	Mass (%)	Components
1	Sugar	~45	Sucrose, glucose, fructose, etc.
2	Moisture	~10	water
3	Crude protein	~10	Amino acids
4	Ash	~5	Potassium, sodium, etc.
5	Other	~30	Non-nitrogenous materials

Note: ~ represents an approximation.

**Table 2 ijerph-19-16472-t002:** The proximate content, maceral composition, and vitrinite reflectance (R_0_) of the coal sample.

Proximate Content (wt%)				Maceral Composition (Vol%)				*R*_0_ (%)
*M* _ad_	*V* _daf_	*A* _ad_	FC_ad_	Vitrinite	Liptinite	inertinite	Sapropelinite	
1.16	26.01	25.79	47.04	72.43	6.83	0.98	19.76	0.60

The shown measures represent air-dried coal. *M*_ad_ represents the moisture content, *A*_ad_ represents ash content, *V*_daf_ represents the volatile matter content (i.e., dry-ash-free), and FC_ad_ represents the fixed carbon content.

## Data Availability

The data presented in this study are available on request from the corresponding author. The data are not publicly available due to privacy or ethics.

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
