# Peer review of "Suppression Characteristics and Mechanism of Molasses Solution on Coal Dust: A Low-Cost and Environment-Friendly Suppression Method in Coal Mines"

_ijerph, 2022, doi:10.3390/ijerph192416472_

Round 1

Reviewer 1 Report

The suppression characteristics and mechanism of molasses solution on coal dust were explored in this manuscript. As a by-production of the sugar industry, molasses has properties of both low-cost and environment-friendly. Exploring the suppression characteristics of molasses on coal dust has a positive effect on the development of efficient and low-cost dust suppressants. With some minor revisions, I believe this manuscript could meet the criteria for publication in IJERPH.

(1) In the fourth paragraph of the Introduction section, the sentence “However, the ability of molasses to suppress coal dust (i.e., especially hydrophobic coal dust), to the best of our knowledge, have not been studied” was suggested to be revised as “However, the ability of molasses to suppress coal dust (especially hydrophobic coal dust) has not been studied”.

(2) In the fourth paragraph of the Introduction section, the last sentence seems to be redundant, and it is suggested that this sentence could be removed.

(3) In the last paragraph of the Introduction section, a brief description of the objectives and significance of the investigation is generally required. However, it is not necessary to make a detailed description of the research method and research process in this section.

(4) In the 3.3 section, there is a punctuation error in the sentence “The viscosity of solution exponentially increases with the increase in the molasses concentration. With a R2 = 0.99;”

Reviewer 2 Report

In this study, the feasibility of preparing a coal dust suppressant using molasses, a byproduct of the sugar industry, was systematically investigated using various experiments. It is an interesting study for developing low-cost and environmentally-friendly coal dust suppressants. The study was recommended published after several minor revisions as follow.

(1) The order number following the authors’ names should be marked using Arabic numerals;

(2) In the penultimate sentence in the Abstract section, it will be more accurate that changed the word “seen” to “discussed”;

(3) To improve the readability of the study, the expression of some sentences in the Introduction section should be simplified and polished;

(4) In Figure 14, it will be better that place the two figures in a line by changing their sizes;

(5)  In the Conclusions section, the first sentence of the third conclusion could be replaced by “Molasses has significant agglutination effect on coal dust”.

Reviewer 3 Report

The paper present a novel method for coal dust suppression in coal mines, which can present high applicability potential in the industry, probably with multiple benefits, such as the mitigation of the health risk for workers and reducing potential explosion risk.

I suggest some minor revisions on the following aspects:

1. English has to be revised. Some typos in the text have been identified (for example "prayed" instead of "sprayed" etc.

2. Table 1 is difficult to interpret in its current form. Try to organize the main constituents and their components on separate, visible rows.

3. In case of the dust suppression efficiency test, it should be described more briefly why 5 m/s wind speed was used in the measurements. Is this an average, or maximum value in coal mines, representing the erosion threshold? Wind friction stress and velocity is highly dependent on particle size distribution, roughness etc.

Overall, I've found your study important and I am looking forward for its future application in mines.
